Intensive pasture management alters the composition and structure of plant-pollinator interactions in Sibiu, Romania

Neacă Ana-Maria 1
Meis Julia 2 3
Knight Tiffany 2 4 5
Rakosy Demetra 4 5 demetra.rakosy@ufz.de
1 Faculty of Biology and Geology, Babeș-Bolyai University , Cluj-Napoca , Romania
2 Institute of Biology, Martin Luther University Halle-Wittenberg , Halle , Germany
3 Helmholtz Centre for Environmental Research–UFZ , Leipzig , Germany
4 Department Species Interaction Ecology, German Centre for Integrative Biodiversity Research (iDiv) Halle-Jena-Leipzig , Leipzig , Germany
5 Department Species Interaction Ecology, Helmholtz Centre for Environmental Research–UFZ , Leipzig , Germany
Silva Daniel
Electronic publication date: 2024 Feb 29
Publication date: 2024
Volume: 12
Electronic Location ID: e16900
Received 2023 Oct 3; Accepted 2024 Jan 16
Copyright: © 2024 Neacă et al.
Copyright year: 2024
Copyright holder: Neacă et al.
License: This is an open access article distributed under the terms of the Creative Commons Attribution License, which permits unrestricted use, distribution, reproduction and adaptation in any medium and for any purpose provided that it is properly attributed. For attribution, the original author(s), title, publication source (PeerJ) and either DOI or URL of the article must be cited.
License URL: https://creativecommons.org/licenses/by/4.0/

Keywords: Grazing, Plant-pollinator interactions, Diversity, Community composition, Networks

Funding: German Centre for Integrative Biodiversity Research (iDiv) Flexpool Support Fund for iDiv scientists 346001151-21 This work was funded by the German Centre for Integrative Biodiversity Research (iDiv) Halle-Jena-Leipzig, through the Flexpool Support Fund for iDiv scientists (Project number: 346001151-21). The funders had no role in study design, data collection and analysis, decision to publish, or preparation of the manuscript.

==============================
Background

Land management change towards intensive grazing has been shown to alter plant and pollinator communities and the structure of plant-pollinator interactions in different ways across the world. Land-use intensification in Eastern Europe is shifting highly diverse, traditionally managed hay meadows towards intensive pastures, but few studies have examined how this influences plant-pollinator networks. We hypothesized that the effects of intensive grazing on networks will depend on how plant communities and their floral traits change.

Methods

We investigated plant and pollinator diversity and composition and the structure of plant-pollinator interactions near Sibiu, Romania at sites that were traditionally managed as hay meadows or intensive pastures. We quantified the identity and abundance of flowering plants, and used transect walks to observe pollinator genera interacting with flowering plant species. We evaluated the effects of management on diversity, composition and several indices of network structure.

Results

Pollinator but not plant diversity declined in pastures and both plant and pollinator taxonomic composition shifted. Functional diversity and composition remained unchanged, with rather specialized flowers having been found to dominate in both hay meadows and pastures. Apis mellifera was found to be the most abundant pollinator. Its foraging preferences played a crucial role in shaping plant-pollinator network structure. Apis mellifera thus preferred the highly abundant Dorycnium herbaceum in hay meadows, leading to hay meadows networks with lower Shannon diversity and interaction evenness. In pastures, however, it preferred less abundant and more generalized flower resources. With pollinators being overall less abundant and more generalized in pastures, we found that niche overlap between plants was higher.

Discussion

With both hay meadows and pastures being dominated by plant species with similar floral traits, shifts in pollinator preferences seem to have driven the observed changes in plant-pollinator interaction networks. We thus conclude that the effects of grazing on pollinators and their interactions are likely to depend on the traits of plant species present in different management types as well as on the effects of grazing on plant community composition. We thereby highlight the need for better understanding how floral abundance shapes pollinator visitation rates and how floral traits may influence this relationship.

Introduction

Global change, such as climate and land use change, have become increasingly evident in every region of the world, threatening biodiversity and critical ecosystem services such as pollination (Allan et al., 2015; Rhodes, 2018; Raven & Wagner, 2021). In recent years, the diversity and abundance of insects in Europe, including wild pollinators, have been generally shown to follow a downward path (Hallmann et al., 2017; Wagner, 2020; Klink et al., 2020). The main mechanisms driving these declines are likely to be, at least in part, linked to changes in floral resource availability determined by land use change (Potts et al., 2010; Weiner et al., 2011; Clough et al., 2014). However, because plants and insects are typically monitored and studied separately, there is insufficient understanding of how particular land management strategies alter pollinators through changing floral resources and the structure of plant-pollinator interactions (Kremen et al., 2007; Steffan-Dewenter & Westphal, 2008; Goulnik et al., 2021). There is thus a need for more case studies in a wide variety of regions.

Grasslands contain a significant amount of the world’s terrestrial biodiversity (Wilson et al., 2012), which is maintained through the complex ecological networks such as those between plants and pollinators (Bastolla et al., 2009). In Europe, there are three main types of grasslands (based on their origin and management): natural, semi-natural, and agriculturally improved (Dengler et al., 2013; Hejcman et al., 2013). Natural grasslands have a highly restricted distribution as their occurrence is limited to particular ecological, edaphic, and climatic conditions (i.e., in alpine regions, steppe) (Hejcman et al., 2013). Most European grasslands are classified as semi-natural, having originated through human intervention (Hejcman et al., 2013). Semi-natural grasslands have thus been created and maintained through low intensity disturbance and biomass removal by human activities for pasture or for hay production aimed at ensuring winter food for livestock (Silva & Kommission, 2008; Hejcman et al., 2013). Agriculturally improved grasslands, on the other hand, are characterized by high intensity use (high livestock load, frequent mowing, addition of fertilizers, etc.), often combined with the addition of agriculturally desirable plant species.

In semi-natural grasslands, decisions about the date and frequency of mowing, the type of grazing animal, intensity of grazing, and other factors have important consequences for their biodiversity and ecosystem services. Semi-natural grasslands managed as traditional hay meadows, are thereby characterized by extensive management practices, defined by relative late mowing dates, one or two cuts combined with late season grazing (or rotational mowing), and no or only limited addition of organic fertilizers (see Babai & Molnár, 2014). Traditional, extensive pastures are usually grazed at low stocking rates for only limited periods within the season (Janišová et al., 2020; Stanciu et al., 2023). These traditional management practices have all but disappeared in most of Western Europe. Eastern Europe, however, still harbors some of the world’s most diverse semi-natural grasslands (Cremene et al., 2005; Dengler et al., 2014); grasslands which have been managed for centuries through low intensity mowing and grazing (Cremene et al., 2005; Kovács-Hostyánszki et al., 2016; Török et al., 2018). However, recent changes in demographic, socio-economic and political factors (Kümmerle et al., 2016; Godde et al., 2018; Nita et al., 2019; Sartorello et al., 2020) are changing grassland management in this region. These factors include an aging rural population, with the younger generations emigrating into urban areas, the loss of the value of hay as food for livestock, as well as financial stimuli from the European Union which have led to a shift in livestock species and their numbers (Kovács-Hostyánszki et al., 2016). Thus, the last 15 to 20 years have seen a shift from smaller herds of sheep, cattle, horses and buffalo to large herds of sheep, as well as abandonment and shifts from hay meadows to intensively-grazed pastures (Kovács-Hostyánszki et al., 2016). European subsidies, together with increasing profitability of sheep exports, have supported the increase in the number of sheep, which has led to more animal grazing in high biodiversity grasslands that had previously been maintained by low-intensity mowing and/or grazing (Roman et al., 2019).

Studies have indicated that traditional mowing and grazing practices are comparably effective in preserving plant and insect diversity and in maintaining their interactions (Lázaro et al., 2016b, 2016a). Nevertheless, the disruption of the management practices which have historically contributed to the formation of hay meadows and pastures is expected to significantly impact their diversity, composition, and ecological function (Bonari et al., 2017). This effect may be even more pronounced when transitioning to more intensive agricultural practices. Intensive grazing is known to alter the structure and composition of plant communities (Kruess & Tscharntke, 2002b; Anderson & Hoffman, 2007; Dumont et al., 2009; Papanikolaou et al., 2011; Hanke et al., 2014; Ganjurjav et al., 2015; Török et al., 2018), resulting in communities dominated by a few species with traits that tolerate trampling, high nutrient content and frequent removal of their vegetative mass (Cingolani, Noy-Meir & Díaz, 2005; Souther et al., 2019). These changes in the taxonomical and functional diversity and composition of plant communities are expected to have a significant effect on the diversity and composition of the pollinator community, as well as on the structure of their interaction networks. The impact of intensive grazing on pollinators and plant-pollinator interaction may thereby, depend on the type and accessibility of floral rewards offered by dominant plant species (Fründ, Linsenmair & Blüthgen, 2010; Fantinato et al., 2018; Goulnik et al., 2021). Thus plant species that dominate in intensively grazed pastures might have flowers with traits that promote easy access to a broad range of pollinators (i.e., generalized flowers) (Goulnik et al., 2021) or they could provide a stronger filter (i.e., more specialized flowers) (Guretzky et al., 2005; Dumont et al., 2009). Only two studies have so far investigated the influence of the functional traits of dominant plant species on the response of pollinators and plant-pollinator interactions to intensive grazing.

In Brazil, Oleques, Vizentin-Bugoni & Overbeck (2019) found that intensive grazing resulted in the dominance of a few Asteraceae species with generalized flowers that support a diverse pollinator community. In contrast, in Poland, intensive grazing resulted in the dominance of Trifolium repens, a species with more specialized flag-flowers that provided a strong filter towards pollinators that could access this resource (Rakosy et al., 2022). Thus, while in Brazil there was little impact of grazing on plant-pollinator networks (Oleques, Vizentin-Bugoni & Overbeck, 2019), in Poland pasture management resulted in less diverse, more specialized and less even interaction networks. Pollinators in these networks showed a high degree of niche overlap, indicating that they increasingly share the same plant resources. These structural changes in networks are expected to influence network stability (e.g., Tylianakis et al., 2010; Bascompte & Scheffer, 2023) and pollinator services (Arceo-Gómez et al., 2020) by increasing the dependance of pollinators on a single or a few plant species and functional types. From a plants perspective increased visitation by pollinators may provide a reproductive advantage by increasing pollen transfer, but it may also have the opposite effect if increased visits are from less efficient pollinators which transfer a higher proportion of heterospecific pollen. While there are a growing number of studies examining the effects of grazing on pollinator communities and on plant-pollinator interactions (Kruess & Tscharntke, 2002a; Cagnolo, Molina & Valladares, 2002; Debano, 2006; Dumont et al., 2009; Jerrentrup et al., 2014; Lázaro et al., 2016b; Opeyemi et al., 2018; Oleques, Vizentin-Bugoni & Overbeck, 2019), few of these focused on how the floral traits of dominant plant species may modulate the shift between management practices in formerly traditionally managed hay meadows, and even fewer are from Eastern Europe (Bennett et al., 2018; Kovács-Hostyánszki et al., 2019).

The goal of the study is to evaluate the effects of grazing on plant and pollinator communities and plant-pollinator networks in an Eastern European study region. We compare grasslands managed extensively as hay meadows to grasslands in the same area that were previously extensive hay meadows but have been converted to intensively-grazed pastures. Our primary focus is exploring how the shift from mowing to intensive grazing affects the taxonomic and functional diversity and composition of plants, and how these changes subsequently impact the diversity and composition of pollinator communities, along with shaping the structure of plant-pollinator interactions. We thereby report the floral types dominating in each land management type, and discuss how differences in pollinator preferences for dominant plant species and floral types might explain the observed changes in network structure. We hypothesized that intensive grazing will reduce the diversity and alter the composition of both plant and pollinator communities, but that the effect plant-pollinator network structure will be more strongly impacted by the dominance of plant species with more specialized floral types, while the dominance of more generalized flower types would mitigate the impact of grazing.

Materials and Methods

Study site

The region chosen for this study is located in the Sibiu Depression, Transylvania, Romania (Fig. 1). The site covers approximately 400 km2 and includes a wide variety of habitats such as deciduous woods, forest edges, mesophilic and meso-xerophilic meadows, pastures, ruderal areas, gardens, and orchards (Schneider-Binder, 1974; Pașcu, 1999). Within this larger area we focused specifically on Gusterița Hill, an area that has been the focus of several studies on the diversity of Hymenoptera and plants (Schneider-Binder, 1974; Pașcu, 1999; Crăciunaș, 2013). We selected 10 sections of the Gușterița Hill, used as either hay meadows (five sections) or intensive pastures (five sections). All of the pasture sections were formerly used either as traditionally managed hay meadows or were lightly grazed pastures that have only recently (in the past 10 to 20 years) transitioned into intensively grazed pastures (based on information obtained from local stakeholders). The grazers are predominantly sheep. Sections of each management type spanned the same altitudinal gradient (500–600 m), shared similar orientation (W–E), were of similar size (approximately 0.8 ha) and were imbedded in a similar landscape, allowing sections managed as pastures to be contrasted with the nearby sections of the hill that are still extensively managed as hay meadows. All sections were expected to be in the flight range of most pollinators, this allowed us to reduce the risk for any shifts in pollinator communities between pastures and hay meadows being due to differences in local pollinators assamblages, rather than a consequence of partitioning due to ressource availability. Permission to sample sites was obtained verbally from the farmers and local stakeholders.

Figure 1 Map of the 10 study sections within the Gușterița Hill area.

Blue points mark the five grazed sections, while orange points mark hay meadow sections. The inserted map (lower left corner) shows the position of the study site within Romania and Europe (inlayed map source https://www.freeworldmaps.net/europe/europe-blank-map-hd.jpg, main map: Google Earth, Maxar Technologies 2023).

Sampling methods

Sampling of plants, pollinators and their interactions within the 10 sections was conducted between June and July 2021, over a period of two weeks, with sampling being repeated three times in every section (at an interval of 3 to 4 days depending on the weather). The sampling period was chosen to coincide with the time period when the majority of flowering plants are in bloom. While peak-season sampling represents only a snapshot into the local plant and pollinator communities and their interactions, the approach has been shown to capture a high proportion of the most important species within a community and to allow relatively precise estimates of network metrics (Hegland et al., 2010). Within each of the 10 sections we established a single 30 × 2 m transect placed at least 30 m from adjacent fringe structures (i.e., forest margins, hedges) and 60 m from other sections.

For each transect, we identified each species of plant that was in flower, and estimated its abundance as the percent cover of their flowers or inflorescences relative to the total area of the transect (Vittoz & Guisan, 2007). We used standardized transect walks to quantify pollinators and plant-pollinator interactions. A transect walk consisted of a single investigator sampling the transect actively for 15 min (i.e., time spent recording observations, capturing and processing insects was not part of the 15-min sampling period). Three investigators thereby rotated between sampling rounds, with each section being sampled by the same three investigators. All data analyses were based on the data pooled by section across the three transect walks. Hymenoptera, Diptera or Lepidoptera which came into contact with the reproductive structures of the plants were considered to be pollinators, even though we acknowledge that insects visiting flowers are not necessarily performing pollination (e.g., Ballantyne, Baldock & Willmer, 2015). Pollinators that could be identified to the genus level on site were recorded but not collected. All others were collected and frozen for future identification. We identified plants to species level (Ciocârlan, 2000) and pollinators to genus using microscopy and insect keys (e.g., Tschorsnig & Herting, 2001; Bartsch et al., 2009; Rakosy, 2013; Michez et al., 2019; Oleques, Vizentin-Bugoni & Overbeck, 2019; Rakosy et al., 2022). In order to assess measures of plant functional diversity and composition, we categorized all flowering species based on their flower shape and structure and the accessibility and type of reward using the classification system developed by Kügler, retrieved from the BiolFlor database (www.biolflor.de, Klotz, Kühn & Durka, 2002).

Data analysis

Differences in sampling completeness can occur even when the sampling effort is the same, and this has implications for interpretation of the results (Chao & Jost, 2012; Chao et al., 2014). We therefore examined whether sampling completeness differed across our management types (hay meadows and pastures). For each site, we estimated the asymptotic richness (Chao1, Macgregor, Evans & Pocock, 2017) of three response variables (plant species, pollinator genera and unique interactions) using the functions “specpool” and “estimateR” from the vegan package (Oksanen et al., 2019) in R (version 4.2). Sampling completeness for each site and response variable is the percent of the asymptotic richness accounted for by the observed richness (Fantinato et al., 2018). We used an unpaired two-sample Wilcoxon test to test whether management types varied in their plant, pollinator and interaction sampling completeness (Vázquez, Chacoff & Cagnolo, 2009).

Measurements of biodiversity can depend on the sample size (i.e., the number of sites) and the metric (i.e., species richness vs. species diversity indices that weight species by their relative abundance) considered (Chao et al., 2014). Thus, we use site-based rarefaction and report two different Hill numbers (q = 0, which corresponds to species richness and q = 2, which corresponds to Simpson’s diversity) to compare hay meadows and pastures. For each site, plant cover and pollinator frequencies were converted to occurrences (presence or absence). We used the “iNEXT” function in the package iNEXT in R (Chao et al., 2014) to interpolate across five sites and extrapolate to double that number for each management type and response variable. Non-overlapping 95% confidence intervals were interpreted as significant differences between management types (Colwell, Mao & Chang, 2004; Chao & Jost, 2012). Plant functional diversity between hay meadows and pastures was, in turn, compared using a Wilcoxon signed-rank test based on the RhaoQ diversity metric calculated using the package FD in R (Laliberté et al., 2014).

To test whether the taxonomic and functional composition of plant and taxonomic composition of pollinator communities differed across the management types, we calculated Bray-Curtis dissimilarity (based on plant species/functional type relative cover and pollinator relative frequency) for all pairwise sites (using the metaMDS function) and used ANOSIM (analysis of similarities) to test whether sites were more dissimilar across than within management types in vegan package (Oksanen et al., 2019) in R. We visualized dissimilarity across sites and management types using nonmetric multidimensional scaling (NMDS). We also visualized the ten most abundant plant and pollinator species across sites in each management type. In order to highlight the dominant functional floral types and their attractiveness to pollinators, we first compared the community weighted mean, calculated using the package FD in R, of floral traits between hay meadows and pastures (Laliberté et al., 2014). We then visualized the relative cover of each flower type as well as the relative number of interactions observed on each flower type across management types. In order to assess the degree to which differences between plant and pollinator communities in hay meadows and pastures are driven by taxonomic turnover or nestedness (i.e., the degree to which communities with fewer taxa area nested subset of richer ones) we calculated the Sørensen dissimilarity index (βsør; Baselga, 2010). βsør was then partitioned into the Simpson dissimilarity index (βsim), reflecting taxonomic turnover, and nestedness (βnes), reflecting the dissimilarity caused by the communities’ nestedness. All three indices were calculated using the betapart package (Baselga & Orme, 2012). We did not assess plant functional turn-over and nestedness, as communities were found not to differ in their functional diversity and composition (see results).

Bipartite plant-pollinator networks were built with plant species and pollinator genera as nodes and interactions between them as edges (Bascompte, 2007; Campbell et al., 2011). We visualized the structure of the hay meadow and pasture networks using the function “plotweb” in the bipartite package in R (Dormann, 2021) and adjusted visual properties using CorelDRAW (version 20.0.0.633). We tested whether network structure differed between hay meadows and pastures, relying on several network level metrics (selected to reflect network diversity and resource partitioning): Shannon diversity, interaction evenness, generality and niche overlap. Shannon diversity reveals the diversity of unique interactions among plants and pollinators and is expected to be linked to higher network stability (Kaiser-Bunbury & Blüthgen, 2015). Interaction evenness provides insights into the distribution of interactions, highlighting shifts in community structure (Blüthgen, 2010; Kaiser-Bunbury & Blüthgen, 2015). Generality quantifies the weighted number of interaction partners of plants or pollinators, with higher generality being linked to higher robustness (Kaiser-Bunbury & Blüthgen, 2015). Niche overlap defines the degree to which co-occurring species in a trophic level share their niche space (e.g., the niche overlap of plants is the degree to which plants share the same pollinator genera) (Kaiser-Bunbury & Blüthgen, 2015). In combination with generality, it provides insights into resource partitioning within plant-pollinator networks. Estimates of all network metrics depend on how many interactions are sampled, and the number of interactions sampled can vary even if the sampling effort is constant. Thus, to more accurately compare the two management types, we utilized the function “boot_networklevel” from the “bootstrapnet” package to generate interaction-based rarefaction curves for network-level indices (Ștefan & Knight, 2020). For each management type we pooled interactions across all sites and then randomly sampled 100 interactions without replacement (with 999 iterations) to create rarefaction curves and 95% confidence intervals for each network metric (Ștefan & Knight, 2020). Apis mellifera played a dominant role within both hay meadow and pasture networks (see results); in order to estimate its impact on the overall network structure we reran all network analyses after removing Apis mellifera from our dataset (Worthy, Acorn & Frost, 2023).

Results

Across all 10 sites, we found 80 unique species of flowering plants in 25 different plant families, 56 genera of pollinators in 24 families, and 220 unique interactions between plant species and pollinator genera. Hay meadow and pastures shared 28 of the 80 plant species, while 25 species were unique to hay meadows and 27 species were unique to pastures. Hay meadows and pastures shared 20 of the 56 insect genera, where 29 genera were only found in hay meadows and seven genera were found only in pastures. For raw data on plant and pollinator communities and plant-pollinator interactions at each site, see Data S1 and S2.

There were no significant differences in sampling completeness across management types for plants (W = 11, p = 0.8294), pollinators (W = 7, p = 0.2963), or interactions (W = 14, p = 0.834). Samping completeness thereby reached 76.8% in hay meadows and 80.93% in pastures for plants; 61.56% in hay meadows and 59.74% in pastures for pollinator genera and 59.02% in hay meadows and 68.22% in pastures for interactions. Sampling completeness values for each of the 10 sections is provided in Table S1.

For plants, taxonomic and functional richness and diversity were not significantly different between management types (Figs. 2A and 2B; functional diversity: WRhaoQ = 4; p = 0,1). In contrast, for pollinators, richness and diversity were significantly higher in hay meadows compared to pastures (Figs. 2C and 2D). Extrapolating to double the number of sampling units, led to a reduction in the detected difference in species richness, but not Shannon diversity (Figs. 2C and 2D).

Figure 2 Site-based rarefaction curves for (A) plant richness, (B) plant Simpson’s diversity, (C) pollinator richness, and (D) pollinator Simpson’s diversity in hay meadows (orange) and pastures (blue).

Solid lines are interpolated values, dashed lines are extrapolated values and shaded colors are 95% confidence intervals.

Taxonomic, but not functional, plant composition was significantly different between land management types (taxonomic composition: R = 0.904, p = 0.0085, Fig. 3A; functional composition: R = 0.224, p = 0.086). In hay meadows, the most common flowering plant species were Dorycnium herbaceum, Medicago falcata and Rhinanthus major (Fig. 4A), and the most common floral types present in the community were flag blossoms and lip flowers (Fig. 5A). In pastures, Trifolium campestre, Achillea millefolium and Trifolium pratense were the most common flowering plant species (Fig. 4B), and flag blossoms and flower heads were the most common floral types (Fig. 5B). Betadiversity values for plant taxonomic diversity reached 0.48 (Sørensen index), with species turn-over contributing most to the differences between plants in hay meadows and pastures (ßSim = 0.47; ßSne = 0.01). Betadiversity values for plant functional diversity were low (ßSOR = 0.17; ßSim = 0.09; ßSne = 0.08).

Figure 3 NMDS analyses for (A) plant species composition and (B) pollinator genera compostion.

Each point represents a site and colors indicate different land management types. Points close to each other have similar bray-curtis similarity in their composition. ANOSIM results are presented in the upper right corner.

Figure 4 The relative abundance of the ten most common plant species in (A) hay meadows and (B) pastures, and the ten most visited plant species in (C) hay meadows and (D) pastures.

The number of visits recieved is illustrated against the relative abundance of plant species in hay meadows (E) and pastures (F). Colours distinguish between land-use types. Light colours demark species which are either abundant but not frequently visited, or frequently visited but less abundant. Asterisks denote species shared between hay meadows and pastures. The larger point size in figures (E) and (F) highlights selected plant species. Images represent the most abundant or the most frequently visited plant species: D. herbaceum (A) and (C) (Image credit: Hectonichus, https://en.wikipedia.org/wiki/Lotus_dorycnium#/media/File:Fabaceae_-_Dorycnium_pentaphyllum-1.JPG,C.C.BY3.0), T. campestre (B) (Image credit: Kevin Thiele, https://www.flickr.com/photos/66951228@N07/6282462089) and Carduus nutans (D) (Image credit: Bernd Haynold, https://en.wikipedia.org/wiki/Carduus_nutans#/media/File:Carduus_nutans_180807.jpg,C.C.BY3.0). All images downloaded from www.wikimedia.org.

Figure 5 Relative abundance of floral functional types in (A) hay meadows and (B) pastures, and the relative abundance of pollinator interactions on floral functional types in (C) hay meadows and (D) pastures.

The number of visits is illustrated against the relative abundance of plant functional types in hay meadows (E) and pastures (F). Asterisks denote floral types shared between hay meadows and pastures, while darker borders denote species which were abundant and were also frequently visited by pollinators. The larger point size in figures (E and F) highlights selected species. Flag blossom image illustrates the most abundant floral functional type as well as the floral fuctional type which received most insect visits (Image credit: Andreas Plank, CC-BY-SA-3.0, https://shorturl.at/grU28).

Pollinator composition was also significantly different between land management types (R = 0.260, p = 0.0289, Fig. 3B). In hay meadows, Apis was highly dominant, whereas in pastures Apis, Sphaerophoria and Lasioglossum were all common pollinators (Fig. 6). Betadiversity of pollinators reached 0.48 (Sørensen index), with species turn-over and nestedness both contributing to the differences between hay meadows and pastures (ßSim = 0.26; ßSne = 0.22). In hay meadows, the plant species most visited by pollinators were D. herbaceum and O. viciifolia (Fig. 4C). While D. herbaceum is common in hay meadows, O. viciifolia had much lower relative abundance (Figs. 4C and 4E). Other abundant flowering plant species, such as M. falcata and R. major received disproportionally few visits (Figs. 4C and 4E). Thus, the flag blossom floral type was visited with high frequency (Figs. 5C and 5E). In pastures, Carduus nutans and Helianthemum nummularium were the plant species most visited by pollinators, despite these plant species being both relatively infrequent in the community (Figs. 4D and 4F). Overall flag blossoms received the most visits, however, relative to their abundance flower head and pollen type flowers were visited more frequently (Figs. 5D and 5F).

Figure 6 The relative abundance of the ten most common pollinator genera in (A) hay meadows and (B) pastures.

Asterisks denote shared genera. Images illustrate the dominant genera: Apis in hay meadows and Apis and Sphaerophoria in pastures (images: Apis by Gilles San Martin under CC BY-SA 2.0; Sphaaerophoria by Ryszard under CC BY-NC 2.0).

Despite the equal sampling effort for hay meadows and pastures, 693 interactions between plants and pollinators were observed in hay meadows and only 250 in pastures. Out of the total number of interactions, 148 unique interactions between plant species and pollinator genera were recorded in hay meadows (Fig. 7), and only 84 in pastures (Fig. 8). Nevertheless, hay meadows were found to have a significantly lower level of interaction diversity (Shannon interaction diversity) and to be less even (interaction evenness) than pastures (Figs. 9A and 9B). Plant species, thereby tended to interact with similar numbers of pollinator genera (i.e., similar generality) across both land management types, but plants in pastures shared pollinators to a significantly larger degree than those in hay meadows (i.e., higher niche overlap) (Figs. 9C and 9D). Pollinators in turn were more generalized in their interactions with plants in pastures (Fig. 9E), but tended to share plant species to a similar degree across land management types (i.e., similar niche overlap) (Fig. 9F). Removing A. mellifera from the networks had the largest effect on the network metrics of hay meadows, leading to an increase of interaction diversity and evenness, an increase in plant generality, but not niche overlap and a slight decrease in insect generality (Fig. S1). In contrast, the structure of plant-pollinator networks in pastures appeared relatively unchanged after the removal of A. mellifera, with the exception of a few pairwise interactions. Thus, Helianthemum nummularium and Trifolium campestre lost the majority of their interactions with the removal of A. mellifera, Carduus nutans remained, however, the most attractive plant within pastures (Figs. S2 and S3).

Figure 7 Interactions between plant species and pollinator genera in hay meadows.

Plants nodes are shown in green and pollinator nodes in red. The number of interactions are illustrated by line thickness and node size. Numbers indicate species which are shared between hay meadows and pastures.

Figure 8 Interactions between plant species and pollinator genera in pastures.

Plants nodes are shown in green and pollinator nodes in red. The number of interactions are illustrated by line thickness and node size. Numbers indicate species which are shared between hay meadows and pastures.

Figure 9 Interaction-based rarefaction curves of network-level metrics comparing hay meadows (orange) and pastures (blue).

(A) Shannon diversity of interactions, (B) interaction evenness, (C) generality of plants, (D) niche overlap of plants, (E) generality of pollinators, (F) niche overlap of pollinators. Solid lines and dotted lines indicate mean values and 95% confidence intervals of rarefaction estimates based on 100 iterations, respectively. The endpoint of the curve corresponds to the same value generated by the ‘networklevel’ function in the bipartite R package (Dormann, 2021).

Discussion

In Eastern Europe, biodiverse, traditionally managed hay meadows are increasingly being converted to intensively grazed pastures (Feurdean et al., 2017; Roman et al., 2019; Nita et al., 2019; Sartorello et al., 2020). There are few studies that examine the consequences this shift in management type and intensity will have for plant and pollinator communities and for the structure of plant-pollinator interactions (Kovács-Hostyánszki et al., 2016; Oleques, Vizentin-Bugoni & Overbeck, 2019; Rakosy et al., 2022). This study aimed to fill this gap with a case study in Sibiu, Romania. Because intensive grazing is known to have a strong filtering effect on plant communities (Rakosy et al., 2022; Dumont et al., 2009), we hypothesized that pastures would be less taxonomically and functionally diverse than hay meadows. We therefore expected pastures to be dominated by one or a few flowering plants species, with the effects of a shift to pasture management on network structure depending on whether the species that dominate in pastures have more generalized or specialized floral structures (Guretzky et al., 2005; Weiner, 2016; Rakosy et al., 2022). We found, unexpectedly, that hay meadows and pastures were similarly diverse in their plant species and functional types. While, management type did shift the species composition of plants, it did not alter the functional types present. Both hay meadows and pastures were thus dominated by plant species with a more specialized flower type. In contrast to plants, we found pollinators to respond more strongly to the two management strategies, with pastures harboring significantly fewer pollinator individuals and genera, which preferentially foraged on plant resources that were not abundant. However, the network structure of pastures was more diverse and more even compared to hay meadows. This effect was to a large extent driven by the strong dominance of A. mellifera and its preferred interaction with D. herbaceum in hay meadows. While plants in pastures attracted a similar diversity of pollinator genera to those in hay meadows, these plants were more likely to share those pollinator genera with other plants in the community. This pattern may contribute to improving network stability, but could have negative consequences for pollinator service if these pollinators deliver mostly heterospecific pollen (Ashman & Arceo-Gómez, 2013).

Traditionally managed hay meadows have been found to be among the most species rich habitats in the world (Dengler et al., 2014). It has been assumed that their conversion to intensive pastures, a trend which is now common in Eastern Europe, would lead to the loss of both plant and pollinator diversity (Rakosy et al., 2022). Our results suggest that the change in land management alters plant communities by reducing the overall abundance of flowering plants and by favoring some plant species over others (i.e., changing composition). This in turn leads to a shift within the associated pollinator communities. Our results are in line with recent ecological syntheses that show that particularly plant diversity at the scale of small local sites is not systematically declining (Dornelas et al., 2013; Vellend et al., 2013; McGill et al., 2015), but that other important aspects of biodiversity, such as the identity of species might be changing (Hillebrand et al., 2018). Such changes in plant composition can have harmful consequences on other trophic levels if, for example, resource availability and accessibility is altered. Thus, pollinator communities may become less diverse, even when no change in plant diversity is detected.

In our study compositional differences in plant communities with similar levels of richness and diversity reflected turnover of plant species within the same floral functional groups. Thus, unlike other two case studies in which grazing and plant functional traits were considered (Oleques, Vizentin-Bugoni & Overbeck, 2019; Rakosy et al., 2022), both hay meadows and pastures in our system were dominated by plant species with specialized floral traits. Specifically, both were dominated by flag flowers which have a closed shape that can be best accessed by bees with short to medium proboscis length. D. herbaceum, which dominates the hay meadows, is able to reproduce vegetatively and can grow in large groups, often outcompeting other meadow species (e.g., Klotz & Kühn, 2002). In the pastures, the annual plant T. campestre dominates the flowering plant community. This species is known to occur in many types of disturbed areas, such as pastures and roadsides.

D. herbaceum was heavily visited by Apis individuals in hay meadows. This might be explained by the managed hives that were in close proximity to one of our hay meadows that was particularly dominated by D. herbaceum, and/or by the preference of Apis individuals for mass-flowering plants such as D. herbaceum (i.e., Rollin et al., 2013). Another plant frequently visited by Apis in hay meadows was O. viciifolia, a Fabaceae with deeper flag blossoms that was far less abundant in the community. The high abundance of interactions between Apis bees and these two flag blossom species in hay meadows explains the low overall diversity and evenness in the plant-pollinator networks. Networks with low interaction diversity and evenness have been shown to be less stable (Kaiser-Bunbury & Blüthgen, 2015). The presence of A. mellifera in hay meadow networks may thus have important negative consequences, particularly as these managed bees might outcompete wild native pollinators by being very efficient at collecting nectar and pollen and by monopolizing floral resources (Shavit, Dafni & Ne’eman, 2009; Valido, Rodríguez-Rodríguez & Jordano, 2019). Considering their potential ecological impact, but also their economic importance (Romania has a long tradition of beekeeping, and honey production is an important industry), it is important to better understand the effects that Apis bees are having on hay meadow networks in Romania in future studies. Such studies will need to focus on either a gradient of A. mellifera abundance or experimentally exclude the species from sites. Simply removing the species from already sampled networks, as has been done in this and other studies (e.g., Worthy, Acorn & Frost, 2023), may account for the impact of A. mellifera on network size and thus network structure, it does, however, not provide insights into the rewiring of interactions expected as a consequence of resource competition between A. mellifera and wild pollinators (Valido, Rodríguez-Rodríguez & Jordano, 2019).

In contrast to hay meadows, pastures were dominated by T. campestre. While sharing the same functional type with D. herbaceum, pollinators visited this abundant flowering species less frequently, preferring two more sparsely flowering species, C. nutans and H. nummularium, instead. These species, with their head and pollen flowers, provide floral resources that are known to attract a broad range of pollinators (Westrich, 2018). Thus, by offering larger amounts of nectar and in particular pollen, these two species may have outcompeted the more abundant T. campestre in attracting floral visitors. While the overall number of pollinators observed and interactions recorded in pastures was 1/3 the number seen with the same sampling effort in hay meadows, the interactions that were observed were more diverse and even. This effect, was however, driven largely by A. mellifera. Removing the species from the networks led to similar levels of interaction diversity and evenness between hay meadows and pastures. Plant generality was also impacted by A. mellifera, becoming significantly lower in pastures with the removal of this species. This is likely because of the high preference of A. mellifera towards D. herbaceum in hay meadows. The pollinators in pastures were more generalized in the plants they visited and the plants shared more of the same pollinator genera (higher niche overlap) in pastures. This can be seen in particular with the relatively common pollinating genera of flies (Sphaerophoria) and bees (Lasioglossum, Andrena). Plants which share a large number of pollinators with their neighbors might receive lower quality pollination services if visiting pollinators deliver mostly heterospecific pollen (Ashman et al., 2020). This is because heterospecific pollen loads can decrease plant reproduction by physically or chemically interfering with ovule fertilization (Morales & Vázquez, 2008; Jakobsson, Lázaro & Totland, 2009; McKinney & Goodell, 2010; Arceo-Gómez et al., 2020). More research is needed linking network structure to plant reproductive output, before the impact of reduced pollination services through grazing can be accurately assessed.

The patterns found in this study on the effects of grazing on plant communities, floral functional traits and plant-pollinator networks differ from those of previous papers (Oleques, Vizentin-Bugoni & Overbeck, 2019; Rakosy et al., 2022). This might be due to many factors, such as differences in the types of grazers present in the pastures, the intensity of the grazing, soil conditions and latitude. Thus, further case studies are needed in order to build towards a more comprehensive understanding on how grazing affects the interaction between plants and their pollinators. Our study highlights the importance of considering whether pollinators are actually using the dominant floral resources. In plant–pollinator networks, plant abundances are known to play an important role in predicting interactions (Vázquez et al., 2009). However, it is possible that the plant that becomes dominant under particular management types is a relatively unattractive resource for the pollinator community. To date we have a very limited understanding on how plant traits which allow plants to be competitive vegetatively under various land management strategies are correlated with traits which make them attractive to pollinators.

Our study would benefit from future research that expands the spatial and temporal grains of investigation. For example, the inclusion of more sites would allow us to disentangle the effects of proximity to Apis managed hives on pollinator community composition and network structure. In addition, we suspect that small scale heterogeneity in climate (soil moisture) and/or priority effects (e.g., that might influence the dominance of D. herbaceum) might also influence plant and pollinator communities and network structure. We note that our study only captures the peak activity of flowering plants and pollinators and not the seasonal variations of their populations. Dominance patterns of species and floral functional types may be temporally dynamic, leading communities to respond differently to land-use change across the year (Kremen et al., 2007; Allan et al., 2015; Hervías-Parejo et al., 2023). An extension of our temporal grain to spring and autumn across multiple years is therefore desirable in the future. Further studies should ideally also aim at higher sampling completeness and taxonomic resolution, especially concerning pollinators. Higher sampling completeness would have allowed better insights into the diversity, composition and interactions of rarer species. However, as abundant species are known to play a key role in driving network structure in particular (Resasco, Chacoff & Vázquez, 2021), our study was able to highlight meaningful distinctions between management types even with incomplete sampling. Ideally plants and pollinators should be resolved to species level, with lower taxonomic resolution, particularly in one of the two partners potentially impacting estimates of network structure (i.e., pollinator generality and plant niche overlap may be overestimated). However, Renaud, Baudry & Bessa‐Gomes (2020) have shown that it is feasible to compare the properties of plant–pollinator interaction networks with a taxonomic resolution lower than the species level. Nevertheless, fully understanding how the functional traits of dominant plant species impact pollinators and their interactions, will require spatially, temporally and taxonomically better resolved studies. Ideally such studies would also include pollen-transport networks, which would facilitate the assessment of true pollination events, rather than visits.

Conclusions

Our study builds towards a more general understanding of how land management influences plant-pollinator interactions through taxonomic and functional changes in the underlying plant and pollinator communities. We found that hay meadows and pastures at a site near Sibiu, Romania, were dominated by plants with specialized flag flowers. In hay meadows these abundant specialists attracted Apis bees whereas in intensely grazed pastures, pollinators preferred less abundant plant species with more generalized floral traits. Furthermore, a much lower density of pollinators was observed in pastures, leading to sparse plant-pollinator networks, which had more diverse and even interactions, more generalized pollinators and plants that are more likely to share pollinators. We conclude that the effects of grazing on plants, pollinators and their interactions will depend on the context, such as the traits of plants present in differently managed sites. Future studies, with extended sampling completeness, spatial and temporal scale and taxonomic resolution will be required to test the generalizability of our results at a European level.

Supplemental Information

Supplemental Information 1 Sampling completeness of plant species, pollinator genera and plant-pollinator interactions for each site.

Sampling completeness is calculated as the percent difference between observed richness in the site and the estimated asymptotic richness of the site.

Supplemental Information 2 Interaction-based rarefaction curves of network-level metrics in which A. mellifera has been removed.

(A) Shannon diversity of interactions, (B) interaction evenness, (C) generality of plants, (D) niche overlap of plants, (E) generality of pollinators and (F) niche overlap of pollinators are compared between hay meadows (orange) and pastures (blue). Solid lines and dotted lines indicate mean values and 95% confidence intervals of rarefaction estimates based on 100 iterations, respectively. The endpoint of the curve corresponds to the same value generated by the ‘networklevel’ function in the bipartite R package (Dormann et al., 2009) .

Supplemental Information 3 Interactions between plant species and pollinator genera in hay meadows after A. mellifera has been removed.

Plants nodes are shown in green and pollinator nodes in red. The number of interactions are illustrated by line thickness and node size.

Supplemental Information 4 Interactions between plant species and pollinator genera in pastures after A. mellifera has been removed.

Plants nodes are shown in green and pollinator nodes in red. The number of interactions are illustrated by line thickness and node size.

Supplemental Information 5 Data on flowering plant species.

The identity of flowering plant species present at each site and the percent cover of their flowers or inflorescences in the 30m by 2m transects. Site = Sampling location; Type = management type; Transect = section number; Date = Sampling date; Visit = first, second or third visit; Family = plant family; Species = plant species following Flora Europea; %corr = relative flower cover for each species (+ and r categories from the Braun-Blanche scale were adjusted to numerical values).

Supplemental Information 6 Data on plant-pollinator interactions.

The pollinator genera and plant species interactions observed at each site, pooled across three 15-minute transects walks. Site = Sampling location; Type = management type; Transect = Number of transect sampled; Date = sampling date; Visit = first, second or third visit; Plant_species = Plant species on which the insect was observed/collected; Order, Family. Genus = taxonomical classification; Individuals = Sum of the number of individuals observed/collected.

We are grateful to László Rákosy for advice on the project and for comments on a previous version of this manuscript, as well as Erika Schneider for information on the study site. We would also like to express our gratitude towards two anonymous reviewers and the editor for their very constructive comments and suggestions.

Additional Information and Declarations

Competing Interests

Author Contributions

Field Study Permissions

Data Availability

The authors declare that they have no competing interests.

Ana-Maria Neacă performed the experiments, analyzed the data, prepared figures and/or tables, authored or reviewed drafts of the article, and approved the final draft.

Julia Meis performed the experiments, authored or reviewed drafts of the article, and approved the final draft.

Tiffany Knight analyzed the data, authored or reviewed drafts of the article, and approved the final draft.

Demetra Rakosy conceived and designed the experiments, performed the experiments, analyzed the data, prepared figures and/or tables, authored or reviewed drafts of the article, and approved the final draft.

The following information was supplied relating to field study approvals (i.e., approving body and any reference numbers):

We have received permission of the local land owners to work on their property, to our knowledge no other permit was required.

The following information was supplied regarding data availability:

The raw data on floral cover and plant-pollinator interactions is available in the Supplemental Files.

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
