# Peer review of "Intensive pasture management alters the composition and structure of plant-pollinator interactions in Sibiu, Romania"

_PeerJ, doi:10.7717/peerj.16900_

## Round 0.1 · original submission · Major Revisions

Dear Dr. Rakosy,

After this first review round, two of the reviewers and I believe your manuscript has merits to be published in PeerJ but major reviews are still needed before accepting your research.

Please consider all issues raised by all three reviewers and prepare a rebuttal letter informing the changes that were and were not accepted by you and your co-authors in the next version of your manuscript.

Sincerely,
Daniel Silva

Reviewer 1 ·

Basic reporting

See my review in Section 4.

Experimental design

See my review in Section 4.

Validity of the findings

See my review in Section 4.

Additional comments

Review of "Intensive pasture management alters the composition and structure of plant-pollinator interactions in Sibiu, Romania” by Neaca et al.

In this study, the authors examined the influence of land management (hay meadows vs. pastures) on plant-pollinator interactions in a specific region of Sibiu, Romania. Key findings revealed that both hay meadows and pastures were dominated by plant species with specialized flag flowers, attracting distinct groups of pollinators. Interestingly, hay meadows hosted a higher density of pollinators and a greater number of interactions between plants and pollinators compared to pastures. Despite the higher abundance of flowering plants in hay meadows, pastures exhibited more diverse and evenly distributed interactions within their plant-pollinator networks. Pollinators in pastures were found to be more generalized in their interactions and shared pollinator genera across plant species to a greater extent than in hay meadows. These results underscore the importance of considering the attractiveness of dominant floral resources to pollinators and the significant impact of management type on both plant and pollinator communities.

The manuscript is well-written; the methods are generally sound, and the results are interesting and well-discussed. Thank you to the authors for the effort they put into collecting and analyzing the data from an understudied system. However, my greatest concern is the lack of a control in your experimental design. The study compares hay meadows and pastures, but it does not include a control group (e.g., natural grasslands without human influence) to better isolate the effects of the specific land management types. Without a natural grassland control, it may be challenging to distinguish between natural variations in insect diversity and plant interactions and those caused by human management (hayfields and grazed pastures). The simplest way to address this would be to explore other existing studies or databases that may have data on insect diversity and plant interactions in natural grasslands of the region. You could use this data for a comparative analysis and interpretation of your results. Regardless of the direction the authors choose to move forward with this study, it is their responsibility to clearly state the limitations of the sampling and experimental design.

I am generally supportive of publication if the suggested changes are implemented. I will describe my suggestions in the order they appear in the text


Abstract

-- L20-22. Please change “"In Eastern Europe, land management of highly-diverse semi-natural grasslands is being shifted towards more intensive grazing..." to "Land management in Eastern Europe is shifting highly diverse semi-natural grasslands towards intensive grazing..."

Introduction

--L57. Rh et al. 2015, is not cited in the list of references.

--L104. Bennett et al. 2018, is not cited in the list of references.

--L114. Change “In Brazil,…” to “For example, in Brazil,…”

--L116, 120. Change “Oleques et al. 2019” to “Oleques, Vizentin-Bugoni & Overbeck, 2019”.

--L120-122 Change this paragraph to: “in Poland pasture management has led to less diverse, more specialized, and less evenly distributed interaction networks among pollinators. They exhibit higher niche overlap with each other, suggesting an increased sharing of the same plant resources”.


--L123. Change “(e.g., (Tylianakis et al., 2010…)” to “(e.g., Tylianakis et al., 2010…)”.

--L130. Change “plant pollinator interactions” to “plant-pollinator interactions”.

--L130-132. "Floral form" and "floral shape" are related terms that are often used interchangeably, but they can carry slightly different nuances depending on the context. "Floral form" typically refers to the overall structure, arrangement, and characteristics of a flower. It encompasses various aspects such as the number and arrangement of petals, sepals, stamens, pistils, and other floral parts. "Floral shape," on the other hand, specifically focuses on the external or visible outline or configuration of a flower. It emphasizes the physical appearance of the flower, including the shape of the petals, overall silhouette, and how the various floral parts are arranged. Please, try to be consistent on these terms through the manuscript.

Materials & Methods

--L154. The study was conducted during a specific (and limited) time period (June to July 2021), capturing the flowering period of plants. Seasonal variations and annual changes in plant and pollinator abundance and behavior might not be fully captured in this limited temporal scope. This should be noted as a significant limitation of the study, emphasizing, above all, the implications on the obtained results.


--L154. Within the sampling period, how often were the data collected? Daily, weekly, bi-weekly? It is necessary to establish the sampling effort.
Even though it is mentioned that the flowering of the sampled plants coincides within the months of sampling (1-2 months can be a very short sampling period), it is necessary to consider that those plants still without flowering could mask the absence of pollinators in some of the sites. Additionally, it is not stated whether the flowering duration is short or long in these species. Providing a justification for this highly limited sampling effort that overlooks the seasonal variation of interacting communities is important.


--L155. What was the distance among the studied sections?


--L155-156. The establishment of a transect 30 m from other sections could be too close. How does the close proximity of these sites potentially affect the results? The flying insects can easily move between sites, right?


--L160. Please specify the schedule used to conduct the observations. This is fundamental especially if some interactions are time-sensitive or not observed within the defined sampling period.


--L161-162. Was it the same or a different observer in all transects or sites? Was this consistent among sites/surveys? The identification of plants and pollinators, especially at the genus level, relies on the expertise and abilities of the observer. Different observers may interpret and identify species differently, potentially introducing observer bias.


--L163-164. Calculating your sampling effort for plant-pollinator interactions amounts to a total of 7 and a half hours throughout your study. Do you consider this sampling effort to be sufficient?


--L223-224. When analyzing the structure of a network, particularly in the context of plant-pollinator interactions, most commonly used network-level metrics includes connectance, nestedness and modularity. Why were none of these metrics included, and on the contrary, it is necessary to justify why the ones used were included? Whatever the response, it is essential to include at least connectance and nestedness to properly address an interest in analyzing the structure of interaction networks.


--L227. Most of these network measures (e.g. Shannon diversity, generality, etc.) are strongly influenced by network size/the number of species in the network. How does this affect your interpretation of the results?


--L258-263. If the sampling completeness percentage is less than 60%, it could be considered as low completeness. This suggests that a significant portion of the pollinator diversity in the area is likely to have been overlooked. This should be discussed as a limitation in the study.

--L273-278. Does 'floral types' refer to different floral shapes? It is important to consistently use the correct terminology throughout the manuscript."


Results

--L265-269. When sampling completeness is high (as in the case of plants), no differences in species diversity (taxonomic and functional richness) are found. However, when sampling completeness is low (as it was for pollinators), then there are differences in richness and diversity. This could support the idea that by increasing their small sampling effort (June-July), these differences are diluted. All of this should be included in the discussion as limitations of the study.

--L306. Change “Pollinator” to “Pollinators”.

--L306-308. It would be interesting to highlight the shared species between networks (Figures 7 and 8); this could be done with letters, numbers, or colors.


Discussion

--L324-326. Given the very short distance between the evaluated transects of both management types, it should be considered that the pasture sections may be being used (despite their poor floral supply) as an alternative foraging site for those insect individuals that are less behavioral competitives. This may explain the low number of individual insects recorded there.


--L373-379. Considering both plant composition and pollinators, what is the main reward that pollinators recorded seek, nectar, pollen, or both? This can help explain differences in visitation intensity between the studied sites.

Reviewer 2 ·

Basic reporting

This study evaluates the impact of land change from meadow to pasture on pollination networks. The research compares plant and pollinator species diversity, along with plant functional diversity. Despite differences in plant composition, taxonomic, and functional diversity remained similar. However, there were disparities in pollinator diversity and composition. Land use change seemed to result in more generalized networks, potentially reducing pollinator efficiency when interacting with plants possessing general traits. The manuscript is well-written, although I have some concerns regarding the methodology.

Experimental design

2. Experimental Design
The experimental design effectively compares plant-pollinator networks between meadow and pasture communities. However, certain crucial information is missing from the methodology section.
(L153) How were sections selected? What were their sizes, and what was the distance between them?
(L171) Why were pollinators classified at the genus level instead of species or morphospecies? While genus classification is acceptable, the rationale behind this choice should be explained. Using species might yield different results; pastures could attract diverse pollinators at the species level, potentially resulting in a less generalized network than it appears.
(L200) There is no prior explanation about the use of plant functional diversity and its significance.
In lines 253-255, the number of unique and shared species shouldn't add up to 57 genera; it actually adds up to 43 (21+15+7=43 genera)4.

Validity of the findings

The findings are valid at the pollinator genus level. In the discussion, it would be pertinent to mention the implications of using genus level to evaluate diversity, composition, and network parameters. Additionally, addressing what results might emerge if species or morphospecies were employed would enhance the discussion.

Additional comments

The manuscript is interesting, well-written, and provides valuable insights into the effects of land use change on various aspects of diversity (taxonomic, functional, network). The results have been interpreted effectively.

Annotated reviews are not available for download in order to protect the identity of reviewers who chose to remain anonymous.

Reviewer 3 ·

Basic reporting

The article "Intensive pasture management alters the composition and structure of plant-pollinator interactions in Sibiu, Romania" by Neaca et al compares sheep grazed pastures to traditional hay meadows in Romania. I appreciate the authors contributing this data in a relatively understudied region and overall I found the manuscript was written clearly and well. I have some concerns/suggestions for the authors to improve clarity.

Some comments:
Throughout the text, it would be helpful if the terminology was either a bit more consistent or better defined. For example, what is the difference between a hay meadow and a grassland (they seem to be used interchangeably)? What is "extensive management" versus "intensive management"? How are the "traditional hay meadows" managed and in what way do the authors think this will have an impact on plant-pollinator interactions? The authors have cited work to support the idea that grazing may or may not be harmful to pollinators, but I don't see any citations referring to traditional hay meadow management or its implications.

Be cautious with the use of "specialised" and "generalised" in the text... it seems these words have more than one meaning here. For example, the "specialised" floral morphology that restricts visitors is not a "specialised" plant in the sense that it still receives many different visitor types. I think it would be better to use "limited accessibiilty" rather than "specialisation" for this just to avoid confusion.

I would avoid strong terminology, like the word "dramatically" which is used a few times throughout.

Experimental design

The study design compared plant-pollinator interactions in five pastures and five hay meadows. They were each sampled along a 30 m transect 3 times for 15 minutes (45 minutes of sampling total for each site). Insects were identified to the genus level on sight or collected and identified in the lab. The plants were identified to species.

I have a few concerns that I think the authors should explicitly address in their paper.
1) This is a relatively small scale observational study. The sites were only sampled three times, in one year, at one location. The authors should be cautious about extrapolating this limited data to the system more broadly.
2) There are different taxonomic levels between the plants and visiting insects. This has potential implications for network structure and the relative generality of the insects versus the plants.
3) There is a significant difference in the abundance of these two categories, but I don't think there was an explicit test of an effect on overall insect abundance. Regardless, can the authors explicitly discuss how this difference in abundance might affect results?
4) The dominance of the honeybee is likely to be driving a lot of these results. The authors state that in the discussion, but it would still be good to see what the results would look like excluding the honeybee (as it is a managed species and hives were close to the study location).
5) The sampling completeness for the insects (even at the genus level) and the interactions was very low, in the 50% range. This seems like it could seriously affect your interpretation of the results. Perhaps discuss how that missing sampling might change things?

Methods details that could be clarified:
what are the sizes of each site?
can you provide more detail on how the percent cover is calculated? Is that the percent cover of flowers relative to all available space or relative to other flowers? Or does this refer to plant cover whether or not it is blooming?
can you provide more detail on how floral traits were measured? The text says the flower structure, accessibility, and reward type were used but not how they were measured on the plants
several of the figures are lacking legends within the figure to explain color and light vs dark
Rather than the bar charts for figures 4 and 5, why not use x/y axes to plot the points against each other? That would indicate whether there was a linear relationship and you could point out outliers that were either more or less attractive than their relative abundance.
In figure 6, why not use total abundance rather than proportional abundance so that it is easier to see how the sites differ from one another. Maybe you could put these bars on the same chart?

Validity of the findings

Overall, I think the authors should be much more cautious about extrapolating from their findings, based on the reasons I listed above. The study was 1) observational, 2) limited in spatial scale, 3) limited in temporal scale, 4) dominated by honeybees that could affect the results, 5) had low sampling completeness of insects and interactions, 6) had different taxonomic resolution between insects and plants.

However, if the authors make these caveats known and are careful about implying their results will apply to other systems, I think that the data were analysed appropriately within the bounds of the study and are valid and interesting.

Additional comments

Other minor comments:
Line 58: I don't think it is correct to state that the abundance and diversity of insects in Europe have drastically declined. It would be better to be more cautious with this statement, e.g. the Hallmann study shows a decrease in insect biomass without showing anything about diversity at one site in Germany and the Klink study shows a decline in terrestrial insect abundance but an increase in freshwater insect abundance in meta-analyses across many regions.
Line 127: what does "hay meadows to grasslands" mean here?
Line 128-130: This is actually more general and not more specific.
Line 131: It says "each" land management type but there are only two categories of land management
Line 268: not sure what doubling the units means here
Line 301: well, there were also fewer interactions overall right? and a low sampling completeness
Line 343-344: was this found here? how does this relate to your results?
Line 346: what does "strong" mean in this context?
Line 354: dominates instead of dominants
In the figures, I think "Asterix" should be "Asterisks" right?

---

## Round 0.2 · Minor Revisions

Dear Dr. Rakosy,

All three reviewers were very positive in this new review round. Apart from minor text improvements suggested by some of the reviewers, I believe you need to clearly establish your hypotheses at the end of your introduction, as pointed out by one of the reviewers. As soon as these minor improvements are implemented, I believe you manuscript will be ready to be accepted for publication.

Sincerely,
Daniel Silva

**Language Note:** The review process has identified that the English language must be improved. PeerJ can provide language editing services - please contact us at [email protected] for pricing (be sure to provide your manuscript number and title). Alternatively, you should make your own arrangements to improve the language quality and provide details in your response letter. – PeerJ Staff

Reviewer 1 ·

Basic reporting

Dear Editor, I wanted to inform you that I have received and thoroughly reviewed the revised manuscript from the authors based on my previous assessment. The authors have demonstrated exceptional responsiveness to the suggestions and critiques outlined in my review. They have diligently addressed all the recommended changes, resulting in a significantly improved and more comprehensive version of the manuscript.

Upon careful evaluation, I am confident in stating that the authors have effectively met all the recommended revisions. At this stage, I find the manuscript ready and suitable for publication in Peerj. I believe no further modifications are necessary, as the manuscript now aligns excellently with the journal's standards and guidelines.

I am pleased to recommend the acceptance of this revised version for publication without reservations.

Experimental design

no comment

Validity of the findings

no comment

Additional comments

no comment

Reviewer 2 ·

Basic reporting

The MS evaluates the effects of management change from conventional hay meadows to intensive grazed pastures, on both plant and pollinators taxonomy diversity and plant functional diversity. It explores how these changes affect the structure of plant-pollinator interaction networks. This study demonstrates that, although plant functionality was similar between the two management types, there was a turnover in plant and pollinator taxa, resulting in greater pollinator generality and niche overlap in the pastures. These changes can have implications for pollinator efficiency in their role as pollinators.
The MS is clear, well-written, and structured. However, it lacks clearly established hypotheses in the introduction. Including hypotheses in the last paragraph of the introduction could enhance the overall clarity and focus of the manuscript.
The references provide valuable feedback and are appropriately used to discuss the results. The figures have a great design and are comprehensive. However, there are some details with figures that need attention:
In Figure 2, a letter 'D' is missing (instead of 'B' for pollinators).
In the legend for Figure 4, the sentence "The larger point size in figures (E) and (F) highlights selected species" needs clarification. Does 'selected' refer to the pollinators?"

Other minor comments:
-A better explanation of the two types of management, hay meadow and intensive pasture, has been provided. However, there are still instances in the manuscript where different terms are used for the intensive pasture: line 28 - intensively grazed by sheep, line 36 - pastures, line 100 - intensively-grazed pastures, line 151 - grassland, line 168 - pasture, line 174 - grazed sections, line 177 - pastures. I recommend that after explaining what a pasture is, the term "pasture" should be consistently used throughout the rest of the manuscript.
-Line 292-293 the sum still doesn't add up (28+25+29= 82 not 80)
-Line 307, indicate that the differences were “significantly higher”

Experimental design

The research questions were well-established and are pertinent in light of the intensification of management in the highly diverse pastures of Eastern Europe, exploring how this intensification can impact plant-pollinator network structure. The methodology employed was deemed adequate, and limitations related to the chosen time window for investigating the study's questions are now addressed. The methodology additionally provides valuable insights into the type of management in the pastures and highlights its differences from that in the hay meadows. Sufficient detail is provided to allow for the replication of the methodology. The analyses are explained in detail and are appropriate for addressing the questions the study aims to answer

Validity of the findings

The results are novel and possess validity, even though the sampling was conducted over a relatively short time period. The authors addressed this limitation by doubling their sampling through extrapolation techniques. Additionally, they employed bootstrapping to compare network parameters, adding robustness to their findings. These results have significant implications for understanding the effects of land-use change on plant-pollinator interactions. The conclusions are well-articulated, directly linked to the research questions, and the authors have been diligent in acknowledging the study's limitations and establishing the validity of the results within their scope

Additional comments

The authors have satisfactorily addressed my comments and those of the other reviewers. For instance, they have provided more detailed revisions of the limitations and have discussed the effect of Apis mellifera in the network, addressing specific comments made during the manuscript review process. Additionally, they have now presented a more robust introduction and methodology, explaining why and how they determined floral functionality

Reviewer 3 ·

Basic reporting

Overall, the authors did a good job of addressing the concerns brought up by the reviewers. Due to some of the fundamental limitations of the study system, I doubt that any more improvements can be made, aside from minor semantic disputes. As such, I am recommending publication with minor revision. The edits introduced some grammatical errors; they will probably be caught by a copy editor (e.g. missing or extra spaces line 76, line 200, line 441, misspelling line 139, 170). I will not try to comprehensively address those.

Experimental design

See above

Validity of the findings

See above

---

## Round 0.3 · accepted · Accept

Dear Dr. Rakosy,

I am pleased to inform you that your manuscript has been formally accepted for publication in PeerJ! Congratulations!

Daniel Silva

Reviewer 1 ·

Basic reporting

After careful consideration I confirm that the authors have diligently addressed all previous suggestions. I believe the manuscript is now in excellent shape and ready for publication. No further changes are suggested at this time. I appreciate the authors' responsiveness and commitment.

Experimental design

no comment

Validity of the findings

no comment

Additional comments

no comment

Reviewer 2 ·

Basic reporting

Authors have incorporated the hypothesis at the end of the Introduction, and other minor comments have been addressed. I believe that the manuscript is now ready for publication in PeerJ

Experimental design

no comment

Validity of the findings

no comment

Additional comments

no comment